# Recognition of a Single Dynamic Gesture with the Segmentation Technique HS-ab and Principle Components Analysis (PCA)

**DOI:** 10.3390/e21111114

**Published:** 2019-11-14

**Authors:** Diana Alejandra Contreras Alejo, Francisco Javier Gallegos Funes

**Affiliations:** Escuela Superior de Ingeniería Mecánica y Eléctrica, Instituto Politécnico Nacional (IPN) Av. IPN s/n, Edificio Z, Acceso 3, 3er Piso, SEPI-Electrónica, Col. Lindavista, Ciudad de México C.P. 07738, Mexico; fgallegosf@ipn.mx

**Keywords:** gesture, principle components analysis (PCA), recognition, technique HS-ab, training

## Abstract

A continuous path performed by the hand in a period of time is considered for the purpose of gesture recognition. Dynamic gestures recognition is a complex topic since it spans from the conventional method of separating the hand from surrounding environment to searching for the fingers and palm. This paper proposes a strategy of hand recognition using a PC webcam, a segmentation technique (HS-ab which means HSV and CIELab color space), pre-processing of images to reduce noise and a classifier such as Principle Components Analysis (PCA) for the detection and tracking of the hand of the user. The results show that the segmentation technique HS-ab and the method PCA are robust in the execution of the system, although there are various conditions such as illumination, speed and precision of the movements. It is for this reason that a suitable extraction and classification of features allows the location of the gesture. The system was tested with the database of the training images and has a 94.74% accuracy.

## 1. Introduction

Human gestures are non-verbal corporal actions that are used to transmit information and to express feelings. These human expressions can be done by different parts of the body; the hand is the most versatile interaction tool, and is effective fpr general use. Gestures have been classified into two categories: static and dynamic [1]. The static gestures refer to a particular pose of the hand without movement, represented by only one image and the dynamic gestures refer to the movements of the hand, represented by several images. The use of the hand gesture has become a research area in Computer Vision (CV) that plays a very important role in the field of Human–Computer Interaction (HCI). The goal of HCI is to improve the interaction between users and computers in such a way that computers are more receptive to the needs of users so that it allows humans to interact with the computer in a natural way without any physical contact with external devices such as mouse, keyboard, touch screen, joystick, etc.

According to Haria [2], the gesture recognition system consists of four main stages: data acquisition, segmentation and pre-processing, feature extraction, and classification.Data Acquisition is responsible for collecting the input data which are the images of the hand captured by RGB cameras, data gloves, Time-Of-Flight camera (ToF) and sensors such as Kinect. Subsequently, Segmentation prepares the image to extract the features in the next stage. Its process consists of dividing the input image into separated regions. The most commonly used technique to determine the regions of interest is the detection of skin color [3], which is used to get the skin pixels in the image. Skin color segmentation can be applied on any color space HSV, YCbCr, YUV, CIELab, etc. Part of the segmentation is Pre-processing that is responsible for image processing subject to different pre-processing techniques such as normalization, noise removal, edge detection, smoothening, and morphological operations followed by different segmentation techniques for separating the foreground from the background. After modeling, the representation of a gesture is based on the appropriate metrics with the Feature Extraction. These features can be the location of the hand, palm, and fingertips; angles of the joints and fingers; fingertip markings; number of stretched fingers; orientation; speed; histograms; any emotional expression; or body movement [4]. The extracted features are an input to a computational model, whose objective is to correctly classify the given gesture or can be stored in the system at the training stage as a model [5]. The final stage is Classification where the command/meaning of the gesture must be clearly identified by the system. A classifier allows associating each input gesture according to similarity with other test gestures (database). The input of this stage is a sequence of data (unobservable) with the parameters of a model that were acquired in the training stage. Some methods that allow classifying the gestures are: Euclidean Distance [6], Support Vector Machine (SVM) [7,8], Clustering [9], Neural Network [10], Fuzzy Systems [11], Hidden Markov Model (HMM) [12,13], Non-Parametric (Nearest neighbor, Minimum Distance to Centroid) [14,15], and Principle Components Analysis (PCA) [16,17]. PCA is a dimensionality reduction technique that can be used to solve body recognition problems. This technique is also known as Hotelling, projection of eigenvalues and eigenvectors, and Karhunen–Loève theorem. The dimensionality reduction helps in the following way: it avoids voluminous calculations and is robust to the noise in the images. PCA transforms the original image data into a subspace set of principal components, of which the first principal component captures the greatest amount of variance among the images, while the second principal component provides the vector of the directions orthogonal to the first principal component and so on. This article focuses on the recognition of manual gestures with a simple application such as painting, using capturing images with a PC camera as data acquisition; application of two spaces colors HSV and CIELab as segmentation; and erosion and dilation as pre-processing. Subsequently, PCA is applied in feature extraction and classification.

## 2. Related Work

It should be noted that all gesture recognition systems consist of several stages such as those mentioned in this work: data acquisition, segmentation and pre-processing, feature extraction and classification. According to the criterion of the authors, each stage uses a technique or method according to the application of the recognition system. For example, there are sign language applications using gesture recognition known as Sign Language Recognition (SLR). Kumar [18] presented a framework to recognize manual signs and finger spellings using Leap motion sensor. This sensor includes the associated Application Programming Interface (API) that allows the capture of the 3D position of the fingertips. Subsequently, the raw data are pre-processed with a zero-mean (z-score) based normalization by computing the standard deviation and mean, and the features extracted are the fingertip positions and three angular direction. Then, these data are classified using SVM to differentiate between manual and finger spelling gestures, and two BLSTM-NN classifiers for the recognition of manual signs and finger-spelling gestures using sequence classification and sequence-transcription based approaches. The database of this proposal is 2240 gestures with the participation of 10 subjects. As a result, the system has an accuracy of 100% by using the SVM classifier but in general terms has an accuracy of 63.57% for both types of gesture classes in real time. Another study on SLR was reported by Kumar [19], who developed a SLR system using a depth-sensor device (Kinect) that, when capturing 3D points of body skeleton, recognizes gestures independently to the signer’s position or rotation with respect to the sensor. Its process is as follows: The signer’s skeleton information is captured through Kinect and the data are processed through affine transformation. This transformation is used to cancel out the effect of signer’s rotation and position while performing the gestures. Then, the features (angular features, velocity and curvature features) from the 3D segmented hands are extracted. Furthermore, gesture recognition is carried out with the HMM classifier in three modes: single-handed (64 Gaussians components and 3 HMM states), double-handed and combined setup (128 Gaussians components, 4 HMM and 5 HMM states). The latter (combined setup) presents an accuracy of 83.77% on occluded gestures. Its database consists of 2700 gestures performed by 10 different signers. As noted in the description of the previous work, a gesture recognition system has various applications according to each project. Each research proposal applies various classifiers according to the best accuracy results; the features extracted depend on the data acquisition sensor and the input variables of the classifier; and the database is customized for the training of the proposed algorithm.

Much work on gesture recognition has focused on detecting the hand as well as its features such as position and size of the hand. Thus, many algorithms have been proposed to identify hand gestures with PCA. PCA is used to reduce the dimension of the data and helps the process of classification the features obtained [20]. Dardas [21] presented a real time system that includes the detection and tracking of the hand with non-uniform backgrounds using a skin segmentation and an algorithm of the hand contour position that implies a subtraction of the face and a base of recognition of gestures of the hand with PCA. Prasad [22] used the PCA to determine the characteristics of all frames in the video sequence and then these are stored in the database. Then, the gestures are classified using an artificial neural network. Sawant [23] proposed a sign language recognition system using the PCA algorithm to compare the features obtained from HSV color space segmentation. After comparing the features of the sign captured with the test database, the minimal Euclidean distance is calculated for the recognition of the sign. On the other hand, Zaki [24] presented sign language recognition that uses PCA as a dimensionality reduction technique and represented the features of the image to provide the orientation of the hand. In [25], PCA characterizes the function of the fingers in a vision-based Chinese SLR system. Aleotti [26] analyzed the use of PCA for automatic recognition of dynamic gestures of the human arm and the imitation of the robots. In [27], PCA is applied for the recognition of hand gestures in order to represent the alphabet. Furthermore, Mandeep [28] focused on a skin color model database and a defined threshold, which they compared with a PCA analysis.

Segmentation is a crucial step in a system because it provides color information for an image. This project proposes a segmentation technique with the use of two HSV and CIELab color spaces. Some studies employ these color spaces but with different applications; for example, Cai [29] presented a system for automatic transfer function design based on visibility distribution and projective method color mapping. These transfer functions play a key role in representing the volume of medical data. The techniques commonly used in the transfer function design are clustering and segmentation, which divide the volume into a set of clusters or segments. The algorithm uses visibility distribution for the design of a transfer function. For the color transfer function, normalized-cut segmentation algorithm on the IGM (Intensity Gradient Magnitude) histogram is used to divide the histogram into several regions. Then, the CIELab color space is applied to map the center of each region in order to assign a unique color to each region by its projected coordinate. Then, the resulting color transfer function is applied to represent the volume of data and generate the visualization result. Another study with a different application to the previous one but making use of color space was reported by Nguyen [30]. This article allows the segmentation of regenerated muscle fibers in mice based on superpixels. Superpixels have numerous applications in computer vision including image segmentation. A method called SLIC-MMED (simple linear iterative clustering on multi-channel microscopy with edge detection), which segments muscle fibers in multi-channel microscopy, is proposed. The green channel is processed to extract the nuclei, while the remaining channels (cyan, yellow and red) are used to generate superpixels according to SLIC-MMED algorithm. This work compares the performance of the proposed algorithm with the original SLIC algorithm, the latter operating on color images in the CIELab color space using the components (*l*, *a*, *b*). However, in SLIC-MMED algorithm, each image channel is used as a component of the feature vector. In the results, the SLIC-MMED algorithm can correctly segment muscle fibers of different sizes with bright and dark regions. The previous works represent that the coloration is widely applied in different fields because the segmentation allows distinguishing different structures for a better compression of the data, and thus to be able to perform a manipulation of data according to the objective.

## 3. Materials and Methods

This section describes the procedure of the proposal of this work.

### 3.1. Data Acquisition

The purpose of this stage is to acquire a sequence of images (video) that subsequently is processed in the next stage. The sequence of images is obtained using a 1.3 megapixel camera from the PC (ASUS Notebook UX32A (ASUSTeK Computer Inc., Taipei, Taiwan)) with different backgrounds and with a variation of lighting conditions.

### 3.2. Segmentation and Pre-Processing

In this stage, the image is prepared so that, later, in the following stage, it is possible to extract the characteristics of the image. Preparation consists of dividing the input image into regions, so that the region of the hand gesture in the image is set aside. The most commonly used technique to determine regions of interest in the image (in this case, hand gestures) is through color detection. A good segmentation will allow detecting the skin color in the image. This work uses a proposed segmentation technique called HS-ab, which consists of transforming the input image into the HSV color space to identify non-skin color pixels with thresholds 0 < H < 0.2 and 0.15 < S < 0.9 in such a way that the pixels in the image with these thresholds are excluded. Subsequently, this same image is converted to the CIELab color space to detect the skin color pixels through the thresholds 142 < a < 225 and 115 < b < 177, so that, finally, the image presents only the skin areas. It should be noted that part of the segmentation process is to build a decision rule that differentiates between the pixels of an image that correspond to skin color and non-skin color. This can be done by a non-parametric modeling such as the histogram-based threshold [31]. This modeling of histogram-based threshold uses an approximate estimate to define the range of a color space corresponding to the desired color that appears in the images. Then, to obtain the thresholds of the chrominance components of the HSV and CIELab color spaces, the images acquired in the SFA Image Database [32] are used. For the analysis of the HSV color space, 40 images are used that do not represent any type of skin color. For the CIELab color space, 40 images of different skin color tones are applied. In the case of non-skin images, each image is transformed to the HSV color space by obtaining the histograms of the H and S channels. Each histogram estimates the minimum and maximum values for component H and component S. After having all the histogram values of the 40 images, the minimum and maximum average value of H and S are obtained. In this way, the thresholds for non-skin color pixels H and S are achieved. On the other hand, in the case of skin images, it is the same procedure described above except that the images are transformed to the CIELab color space resulting in the minimum and maximum average values of components a and b.

Later, the image is converted to binary so that morphological operations can be applied. The two basic morphological operations are erosion and dilation, which are used to eliminate the noise in the images and expand and reduce the structure of the objects for their detection. Erosion has the effect of thinning the area of the image while dilation has the effect of widening the region of interest (skin regions) of the image. The PCA has two stages: training and testing. In the training stage, the Eigen space is established using hand gesture training images and, at the same time, these images are mapped in the Eigen space. At the testing stage, when the image is approved (that is, it is mapped to Eigen space), it is classified using a distance classifier. The training stage is then presented in Section 3.3 and the testing stage in Section 3.4.

### 3.3. Feature Extraction

Step 1: Create a set of hand gesture training images and get the database. The database contains training images (M) of dimensions N×N: $I_{1}$, $I_{2}$, …, $I_{M}$.

Step 2: Represent each image $I_{i}$ in a vector $Z_{i}$ where 1≤i≤M with dimension $N^{2}$.

Steps 3 and 4 allow normalizing the vectors of the images of gestures of the hand. This normalization eliminates common characteristics of the images of gestures, in such a way that every image vector represents to the only characteristic.

Step 3: Calculate the average of the vector of the image:
(1)$$\Psi =\frac{1}{M}\sum_{i=1}^{M}Z_{i}$$


Step 4: Get the difference image by subtracting the vector from the average image of each vector from the gesture training image:
(2)$$\Phi_{i}=Z_{i}-\Psi$$


Thus, $\Phi_{i}$ is the normalization of the vectors of the gesture images, $Z_{i}$ is a vector of the gesture image and Ψ is a vector of the average image.

The following steps are for calculating the Eigen vectors but first the covariance matrix *C* must be calculated.

Step 5: Calculate the covariance matrix *C* with dimensions $N^{2}\times N^{2}$:
(3)$$C=\frac{1}{M}\sum_{n=1}^{M}\Phi_{n}\Phi_{n}^{T}=AA^{T}$$
where $A=[\Phi_{1},\Phi_{2},\Phi_{3},\ldots ,\Phi_{M}]$ is a column of matrix with each gesture normalized with dimensions $N^{2}\times M$ where $N^{2}$ is the number of rows and *M* is the number of columns.

The dimensions of $AA^{T}$$\left[N^{2}\times N^{2}\right]$ are very large, therefore it generates $N^{2}$ Eigen vectors of big size; the dimensions are as big as the gesture’s own image. Therefore, there is a need to reduce the dimensionality with a reduced dimensionality covariance matrix. This can be done by calculating the covariance matrix in the following way:
(4)$$C=A^{T}A$$


Step 6: Calculate the Eigen vectors $\left(u_{i}\right)$ of $A^{T}A$ with dimensions M×M, which has the same Eigen vectors and Eigen values of the covariance matrix $AA^{T}$. Then, to calculate the $u_{i}$, it has to be that the total number of rows in the covariance matrix gives the total number of Eigen vectors. That is, the total number of Eigen vectors =M.

Step 7: Calculate the best *M* Eigen vectors of $AA^{T}$ with the following equation:
(5)$$v_{i}=Au_{i}$$
where $v_{i}$ is the upper Eigen vector in dimensional space while $u_{i}$ is the lower Eigen vector in dimensional space.

Step 8: Take only the Eigen vectors (V) corresponding to the largest Eigen values (V).

To make the representation of the database of the training images using the Eigen vectors is when each gesture is projected in the space. It is for this reason that the projection of the gesture is calculated through the weight of each training image; therefore, the weight is calculated as follows:
(6)$$w_{j}=v_{j}^{T}\cdot \left(Z_{i}-\Psi \right)$$
where j=1,2,3,…,M.

As the weight vector is represented as:
(7)$$\mu ={\left[w_{1},w_{2},w_{3},\ldots ,w_{M}\right]}^{T}$$
each image in the training database is represented by the weight vector in the following way:
(8)$$\mu_{i}={\left[w_{1}^{i},w_{2}^{i},w_{3}^{i},\ldots ,w_{M}^{i},\right]}^{T}$$


Thus, for each training image, the weight vectors are calculated and stored.

### 3.4. Classification

Step 1: Then, the test stage is performed. For this, a test image is required (image with any gesture of the hand) (Z) from which its weight $\left(w_{i}\right)$ is calculated through the multiplication of Eigen vector $\left(v_{i}\right)$ with the image difference; this is described in the following equation:
(9)$$w_{i}=v_{i}^{T}\cdot \left(Z-\Psi \right)$$


In the same way, the test image is represented by the weight vector and is calculated as:
(10)$$\mu ={\left[w_{1},w_{2},w_{3},\ldots ,w_{M}\right]}^{T}$$


Step 2: Calculate the Euclidean distance between the weight vector of the test image (μ) and the weight vectors of the training image database $\left(\mu_{j}\right)$.
(11)$$e_{d}=min∥\mu -\mu_{j}∥$$


Finally, a type of gesture is determined by presenting a minimum Euclidean distance with respect to the test gesture (which is the test image). Thus, to classify the input image (test image) with the correct match of the training database the k-nearest neighbor algorithm (also named K-Nearest Neighbors (KNN) Classifier) is used. Therefore, KNN algorithm is used in this project to classify a test image based on the similarity measure of a set of training image database. KNN algorithm calculates the distance between each unknown instance (test image) and all known instances (training image), and the k-nearest neighbor in known instance sets are selected. The nearest neighbors can be determined by calculating based on distinct distance metrics, such as Euclidean distance [33]. This distance metric is used in this work where the classification is performed by finding the nearest Euclidean distance of the transformed weights of the test image, i.e., a neighbor is deemed nearest if it has the smallest distance, in the Euclidean sense. The choice of the number of neighbors (*k*) is discretionary and up to the choice of the users. On this occasion, *k* = 1 is chosen, because the algorithm classifies the class of the nearest neighbor. Typically, the test image is classified according to the labels of its k-nearest neighbors by a majority of votes. Then, if *k* = 1, the object (test image) is classified as the class of the object (some training image) closest to it and also presents greater precision in having good results. In this work, *k* > 1 is not chosen because with more *k* points the algorithm takes longer to process and receive a response, while a quick response is desired because the gesture detection is in real time. In addition, the training image closest to the image test is searched, thus more *k* points are not considered, only the nearest one that appears in *k* = 1. Figure 1 shows the operation of the k-nearest neighbor algorithm.

The steps described above, namely data acquisition, segmentation and pre-processing, feature extraction and classification, are described in Figure 2, where the hand gesture recognition processing is observed.

## 4. Results

The PCA algorithm requires two databases, the training database and the testing database. In the first database, each gesture of the training database is represented as a column vector, that is, the values of each column that forms a matrix represent the pixels of the gesture image. Subsequently, the algorithm calculates the average of this matrix to normalize the gesture vectors. Then, the Eigen vectors of a covariance matrix is obtained. Finally, each gesture is projected to the corresponding gesture space when the Eigen vector matrix is multiplied by each of the gesture vectors. At the testing database, a gesture of a subject (which is a test gesture) in the same way is represented as a column vector that is subsequently normalized with respect to the average gesture. Then, the algorithm projects the test gesture into the gesture space using the Eigen vector matrix. Next, the Euclidean distance is calculated between this projection and each of the known projections of the database in the training stage, so that the algorithm selects the minimum value of these comparisons. This indicates that the algorithm recognizes a gesture from the training database with respect to the test gesture. It should be noted that at this stage the testing database is used.

Training and testing data are sets of manual gesture images. The training database is for the PCA algorithm training, while the testing database is for testing the flexibility and adaptability of the proposed gesture recognition algorithm. Another difference of both databases is that the images in each database are different, i.e., the set of gestures in the training database belong to some test subjects while the gestural images of the testing database are acquired from other test subjects who are different from those who participated for the training database. Therefore, there are no images from the training and testing database belonging to the same person as each image in both databases belongs to different people.

Training database: This database contains images of gestures performed by 20 people (there are 10 female subjects and 10 male subjects) and each person develops five gestures. Therefore, there are a total of 100 images of manual gestures for each hand posture. The images are captured using a PC camera. The database consists of three hand postures: index, V sign and fist under a uniform background (see Figure 3). These last two gestures are attached to experience the testing database of other studies [34,35] with the algorithm of this work. Marcel [34] used a neural network model to recognize six hand postures where an analysis of face location and body anthropometry is used through space discretization. On the other hand, Maqueda [35] presented a human computer interface to control an application by executing manual gestures using a binary Support Vector Machine classifier and Local Binary Patterns as feature vectors.

Testing database: The database of the V sign and fist hand posture was obtained from [34,35] with different lighting conditions. The database of Marcel [34] contains thousands of various images with both uniform and complex backgrounds; in this case, the images with a uniform background were used. In [35], the database contains 30 video sequences performed by six different individuals. The video sequence was recorded on a non-uniform background and each video sequence was spatially and temporally segmented, so the training samples of each gesture are cropped video sequences, whose frames are regions of image that contain only the poses of the hand. In each database, 100 images of manual gestures were acquired. Furthermore, for the index hand posture database, the same procedure described above of the training database was performed, i.e., there are 100 images with a uniform background of 20 different people. Some images of the testing database are presented in Figure 3.

It should be clarified that the images of the three hand postures, namely index, V sign and fist, that belong to the training database and the images of the index posture of the testing database were captured on a PC camera by 20 different subjects. However, testing the operation of the system proposed in this work involves 20 people who are different from the subjects who participated in the capture of the gestural images for the database. Figure 4 shows some results of the application of the recognition system of the three gestures (index, sign V and fist) in the hand with a paint simulation. This hand gesture recognition algorithm was implemented on the Microsoft Visual Studio 2017 Express platform with the SDK version 3.0 library.

On the other hand, in Table 1, the efficiency of the gesture recognition system is shown considering the equations for the calculation of recall (Equation (Equation 12)), precision (Equation (Equation 13)) and prevalence (Equation (Equation 14)). The system was tested by 20 different subjects in real time for the detection of index, V sign and fist gestures. The results presented in Table 1 show that Maqueda [35] presented a lower percentage than Marcel [34], which could be because the images in the training database have non-uniform backgrounds compared to Marcel [34].
(12)$$Recall=\frac{TruePositive}{TruePositive+FalseNegative}$$
(13)$$Precision=\frac{TruePositive}{TruePositive+FalsePositive}$$
(14)$$Prevalence=\frac{TruePositive+FalseNegative}{TotalNumberOfGestures}$$


The average execution time (this is the maximum time to recognize a gesture) of the proposed hand gesture recognition system for each gesture capture in real time is 2.6 ms. This test 3 we performed on a computer with an Intel Core 1.80 GHz processor, 4 GB RAM, a 1.3 MP camera and a resolution of 1280 × 720 pixels.

Considering the results of recall, precision and prevalence of this work, a comparison was made with other methods. Some methods use the PCA algorithm for analysis in the database applied in a gesture recognition system. Kumar [36] presented a system that recognizes specific hand gestures to transmit information in a device control. For this, the PCA is used for manual gesture recognition and the Sheffield KInect Gesture (SKIG) dataset are extracted to create a training set. The training set consists of 10 categories of hand gestures: circle (clockwise), triangle (anti-clockwise), up–down, right–left, wave “Z”, cross, come here, turn around and pat. These gestures are performed with three hand postures, namely fist, index and flat, under three different backgrounds, namely wooden board, plain paper and paper with characters. Table 2 shows the results of recall, precision and prevalence of 10 gesture tests in the system. Mandeep [28] focused on a hand gesture recognition system based on skin color modeling in the YCbCr color space. Then, the image is divided into foreground and background through a defined threshold. Finally, a template based matching technique is developed using PCA. The datasets are in two forms: controlled and uncontrolled. The first has images with a similar background taken from several people. The second presents images with different lighting conditions. The results obtained using both databases to test the system are shown in Table 2.

As shown in Table 2, Mandeep [28], Kumar [36] presented very good results because their training and test databases are very similar (see Figure 5b). In addition, the backgrounds of the images are very uniform and controlled under ideal conditions (see Figure 5c), which allows the database to be more efficient and therefore provide maximum precision. It is for this reason that the percentages of this work tend to be low in the majority of cases: the images in the training and testing databases were acquired by different people (see Figure 5d).

As shown in Figure 5, Kumar [36] only compared identical images and did not test with other images, which is why they offered a precision of 100%. Mandeep [28] contained images with good conditions considering the background and lighting, thus they also offered a precision of 100% but, when testing with images with low illumination, the precision reduced to 96.97%. however, they ded not present results with different gestures and under different conditions as carried out in this project. This project was tested with different images of gestures of various people in real time and with non-uniform backgrounds under different lighting conditions.

Finally, some tests are presented of the system for experimental purposes. First, a comparison of equal images was made in the training and testing databases. Only for this test, some images of the index gesture of the training and testing databases belonged to the same person. This does not mean that the databases of this system have images of the same person, and this test was only done to know the operation of the Euclidean distance with the same, similar and different images. It was observed that, when a test gesture that already exists in the training database is entered, the projection distance (i.e., the Euclidean distance) tends to be almost or completely zero; therefore, there is an encounter of the same gesture. However, when a different test gesture from the images in the training database is shown, the Euclidean distances presented may be far or close to zero. Thus, when comparing all Euclidean distances, the smallest projection distance is chosen because it corresponds to the test gesture. The above is presented in Figure 6. Figure 6a shows that the test image is the same as the training image so it presents a Euclidean distance of 0.0 × 10^10^. Figure 6b shows the test gesture is similar to the training gesture but is not the same, with a Euclidean distance of 0.6 × 10^10^. Figure 6c shows a different test gesture to the training gesture with a projection distance of 1.3 × 10^10^.

The second test was to apply the training database of this work in an algorithm from another study. Dakkata [37] presented a prototype system to recognize the gesture of deaf people based on ASL (American Sign Language) in order to be able to communicate with people. The algorithm consists of the following steps: skin detection, image segmentation, image filtering, and cross-correlation method. This project uses a PC camera to capture the gestures of the person and, after the image processing, the system applies the gray algorithm for illumination compensation in that picture. The database of Dakkata [37] contains 160 images with some types of gestures that represent the ASL with 10 images for each gesture. The training database of the proposed project presents three hand postures, which also belong to ASL. Then, 100 images of each posture (index, V sign and fist) were implemented in the algorithm database of Dakkata [37]. It should be noted that the algorithm [37] was implemented in MATLAB 2019. The results are shown in Figure 7.

## 5. Discussion

A novel approach to dynamic gesture recognition is using the HS-ab technique and PCA for segmentation, feature extraction and classification. First, in recognition, it requires locating some part of the body and eliminating the backgrounds of the image captured by the camera; for this, segmentation is required. This segmentation is crucial because it isolates the relevant data from the background of the image. Then, for the detection of the body parts, a segmentation of the skin color is performed. An important decision for skin color segmentation is the selection of the color space to be used in the algorithm. There are several color spaces; in this case, it focuses on the color spaces with better segmentation in the skin color and on those dependent on the chromaticity components, that is, without the luminance components, because, it presents better results with the variations of lighting found in the environment. The appropriate color spaces with the above characteristics are HSV, YCbCr and CIELab [38]. That is why this work uses the HSV and CIELab color spaces. On the other hand, PCA is recognized in the literature as an accurate method, which is reflected in the statistical analysis of this work. Therefore, PCA is feasible in the use of dynamic gesture recognition. There are other methods for gesture classification, but this method is chosen for its reduction in dimensionality to what leads to a low computational cost and then better response in real time.

To make the proposed system more accurate in gesture recognition, it is considered as future work to apply improved methods of gesture classifiers such is the case of Nguyen [39]. This paper presents a method for multi-user biometric recognition in a prototype gesture-based surgical data access system using a Kinect sensor to capture depth images of a user’s palm. In this literature, a new algorithm is named Centroid Displacement-based K-Nearest Neighbors Classification (CDNN) using the Euclidean distance. This improved algorithm assigns user labels to each presented palm based on the centroid displacement of the classes present in the neighboring points. The accuracy of this algorithm on hands with gloves is 96.24% while the KNN algorithm presents an accuracy of 90.22%. Thus, the CDNN algorithm is more accurate than other classifiers for palm recognition. The objective of applying other modified methods that have presented good responses is to make this system more feasible and more precise in gesture recognition.

## 6. Conclusions

In this paper, hand gesture recognition is developed using the segmentation technique HS-ab and PCA. The system is tested with different environmental conditions such as light and background. The system shows 94.74% recall, 94.74% precision and 95% prevalence. This indicates that the test images match the images in the training database. Therefore, the system is suitable for the recognition of the dynamic hand gesture in Figure 2. Then, the proposed algorithm indicates that, when the hand gesture is entered (which already exists in the training image database), the projection distance calculated for that specific gesture is minimal and therefore its coincidence is excellent. However, when an unknown gesture is entered, the system does not respond. In the future, the training database must be updated to support more than one gesture, and other simulations such as collecting objects, deleting, and moving objects, among others can be developed.

## Figures and Tables

**Figure 1 entropy-21-01114-f001:**
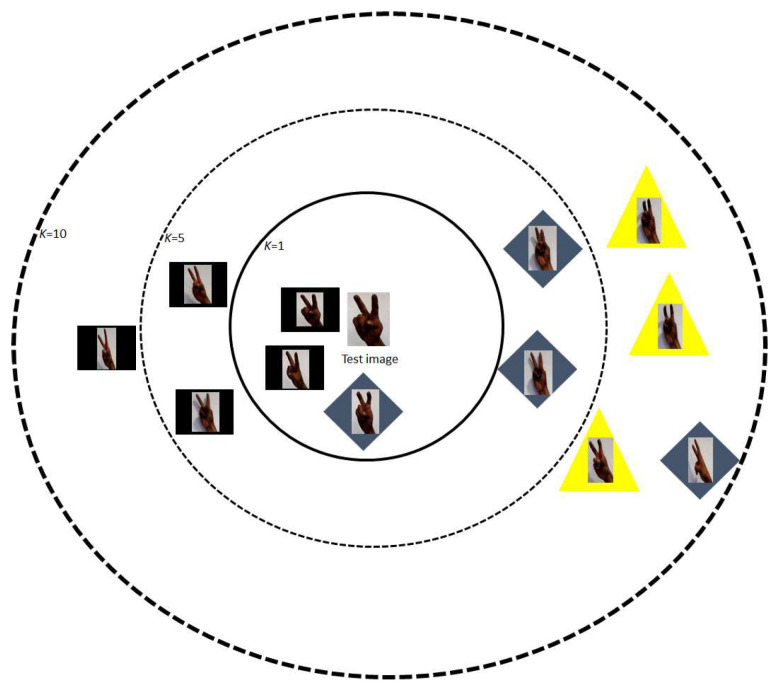
Example of the KNN Classifier with manual gestures. The test image can be a rectangle, a diamond or a triangle depending on the value of *k*. If *k* = 1 (solid line circle), this is assigned to the rectangle; if *k* = 5 (dashed line circle), this is determined to the diamond; and, if *k* = 10 (solid dashed line circle), it belongs to the triangle. The algorithm only considers the object (rectangle, a diamond or a triangle) nearest to the image test of *k* = 1.

**Figure 2 entropy-21-01114-f002:**
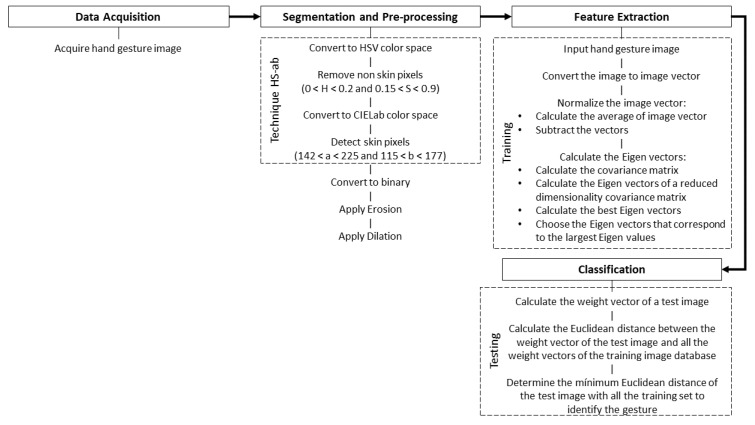
Block diagram of the hand recognition proposal.

**Figure 3 entropy-21-01114-f003:**
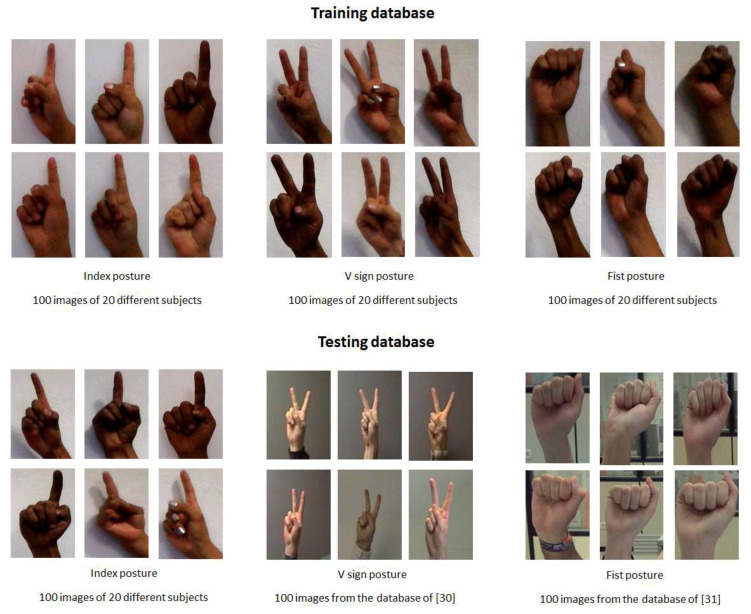
Gestural images of the training and testing database.

**Figure 4 entropy-21-01114-f004:**
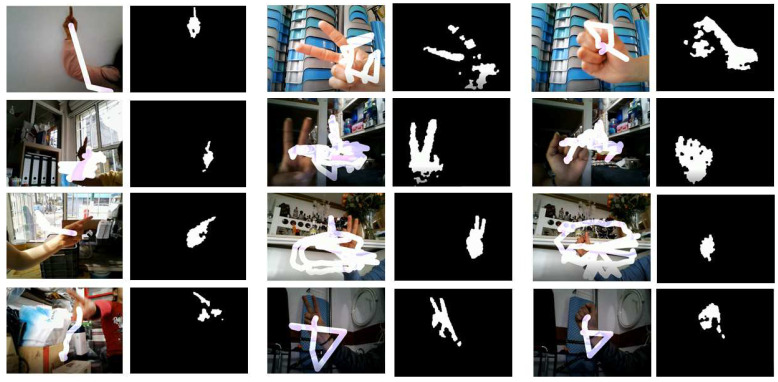
Detection of three types of manual gestures with painting simulation and with different backgrounds. Some images were segmented with the technique HS-ab and the other images were captured from the camera in real time.

**Figure 5 entropy-21-01114-f005:**
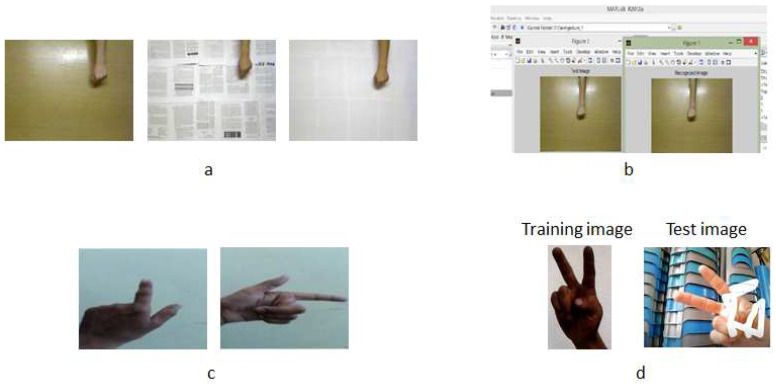
Images of the database in Table 2: (**a**) the training images with different backgrounds but under the same conditions of the database of Kumar [36]; (**b**) the gesture recognition with similar images , however there is no evidence of other gestures in different backgrounds such as in this work; (**c**) some images from the training database of Mandeep [28], of which it is presented in ideal conditions; and (**d**) one of the images of the training database is presented compared to an test image in real time of this project.

**Figure 6 entropy-21-01114-f006:**
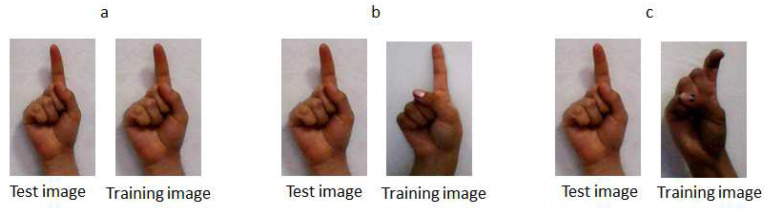
Comparison of Euclidean distances in different cases with the test images.

**Figure 7 entropy-21-01114-f007:**
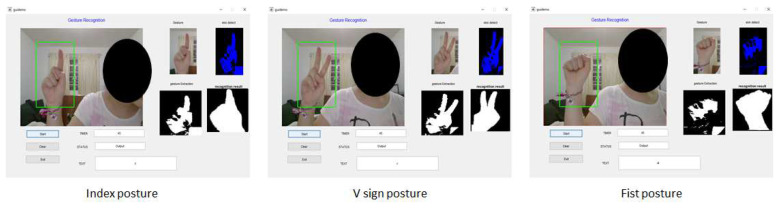
Results of the algorithm [37] using the training database in Figure 3. When the subject performs any of the index, V sign and fist postures, the program of Dakkata [37] detects the type of gesture, the training image and displays the letter that belongs to the gesture.

**Table 1 entropy-21-01114-t001:** Statistical analysis of the proposed hand recognition system.

Parameter	Value	Value [34]	Value [35]
Total Number of Gestures (Total Tested)	20	20	20
True Positive	18	18	17
True Negative	0	0	0
False Positive	1	1	1
False Negative	1	1	2
Recall	94.74%	94.74%	89.47%
Precision	94.74%	94.74%	94.44%
Prevalence	95%	95%	95%

**Table 2 entropy-21-01114-t002:** Test results of the gesture recognition system of various methods.

Reference	Total Tested	Recall (%)	Precision (%)	Prevalence (%)
On this project	20	94.74	94.74	95
[36]	10	100	100	100
[28]	30 (controlled background)	100	100	100
[28]	35 (uncontrolled background)	96.97	94.12	94.29

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
