# Peer review of "Recognition of a Single Dynamic Gesture with the Segmentation Technique HS-ab and Principle Components Analysis (PCA)"

_entropy, 2019, doi:10.3390/e21111114_

Round 1
Reviewer 1 Report
The work is nice. I have following comments.
Only 20 gestures are taken. Can not be generalized. How many users were enrolled for data collection. It is not clear. If few candidates were enrolled, the system could be biased. Can not be generalized. No comparison made with previous studies. Major revision is required.Author Response
To whom it May concern,
"Please see the attachment"
Thank you for your time and for your review.
Best regards,
Authors

Reviewer 2 Report
The authors should take into account the following suggestions in order to improve the manuscript:
Numbering for Sections should start from 1, not 0 In Section Introduction, paragraph 2, the authors listed a lot of different methods for gesture classification including Euclidean Distance, Support Vector Machine (SVM), Clustering, Neural Network, Fuzzy Systems, Hidden Markov Model (HMM), Non-Parametric (Nearest neighbor, Minimum Distance to Centroid), Principle Components Analysis (PCA). However, those were referred to only 2 papers ([2, 6]). They are both just conference papers published in low ranked conferences. I suggest to remove [2] and include other missing references in the same topic, for example, 10.1145/2542050.2542060 (DOI). In Section 2.2, how did you choose the thresholds for a and b? The classification method used in this paper is actually the k-NN algorithm with k=1. Have you implemented the method as the k-NN algorithm? Did you try it with k>1? Please describe the datasets used in the experiments in more detailed, especially the way to split the data into the training and test sets. Is there any image in the test sets belong to the same person having data in the training sets? Some terms (e.g., HSV, YCbCr and CIELab) should be included in Abbreviations. Comparison should be done, either with other classification methods or other frameworks using the same datasets.A major revision is needed for this paper.
Author Response
To whom it May concern,
"Please see the attachment"
Thank you for your time and for your review.
Best regards,
Authors

Round 2
Reviewer 1 Report
I think the authors have made an extensive revision. The paper could be accepted now. I suggest that the authors discuss these papers as well.
A position and rotation invariant framework for sign language recognition (SLR) using Kinect
P Kumar, R Saini, PP Roy, DP Dogra Multimedia Tools and Applications 77 (7), 8823-8846 Real-time recognition of sign language gestures and air-writing using leap motion P Kumar, R Saini, SK Behera, DP Dogra, PP Roy 2017 Fifteenth IAPR International Conference on Machine Vision Applications Thank you.Author Response
To whom it May concern,
"Please see the attachment."
Thank you for your attention and for having taken time for the revision of this manuscript.
Best regards,
Authors

Reviewer 2 Report
The revised version is better than the first submission. The authors may want to consider the followings for further improvement of the paper.
Segmentation or clustering on color spaces HSV and CIELab has been used in some papers in other fields, e.g. 10.1016/j.compmedimag.2013.08.008 and 10.1186/s12918-016-0372-2 (DOIs). Those papers can be added to Section 1 or 2 as related papers. It is suggested to compare the proposed method with an enhance k-NN algorithm (DOI: 10.1109/THMS.2015.2453203) which was successfully applied in a similar application (using k=1). If comparison is not possible, discussion about a future work should be included. Please make sure that all the subjects in the test set have no images in the training set.Another revision is needed for this paper.
Author Response
To whom it May concern,
"Please see the attachment."
Thank you for your attention and for having taken time for the revision of this manuscript.
Best regards,
Authors
